# Spontaneous Osseous Fusion after Remodeling Therapy for Chronic Atlantoaxial Rotatory Fixation and Recovery Mechanism of Rotatory Range of Motion of the Cervical Spine

**DOI:** 10.3390/jcm11061504

**Published:** 2022-03-09

**Authors:** Kazuya Kitamura, Ken Ishii, Narihito Nagoshi, Kazuhiro Chiba, Morio Matsumoto, Masaya Nakamura, Kota Watanabe

**Affiliations:** 1Department of Orthopaedic Surgery, National Defense Medical College, Saitama 359-8513, Japan; kazuya@mre.biglobe.ne.jp (K.K.); kchiba@ndmc.ac.jp (K.C.); 2Department of Orthopaedic Surgery, School of Medicine, International University of Health and Welfare (IUHW), Chiba 286-8520, Japan; 3Department of Orthopaedic Surgery, International University of Health and Welfare (IUHW) Narita Hospital, Chiba 286-8520, Japan; 4Department of Orthopaedic Surgery, International University of Health and Welfare (IUHW) Mita Hospital, Tokyo 108-8329, Japan; 5Department of Orthopaedic Surgery, Keio University School of Medicine, Tokyo 160-8582, Japan; nagoshi@keio.jp (N.N.); morio@a5.keio.jp (M.M.); masa@keio.jp (M.N.)

**Keywords:** atlantoaxial rotatory fixation (AARF), chronic AARF, remodeling therapy, facet deformity, spontaneous osseous fusion

## Abstract

We aimed to investigate the risk factors of spontaneous osseous fusion (SOF) of the atlantoaxial joint after closed reduction under general anesthesia followed by halo fixation (remodeling therapy) for chronic atlantoaxial rotatory fixation, and to elucidate the recovery mechanism of the rotatory range of motion (ROM) after halo removal. Twelve patients who underwent remodeling therapy were retrospectively reviewed. Five patients with SOF were categorized as the fusion group and seven patients without SOF as the non-fusion group. Three dimensional CT was used to detect direct osseous contact (DOC) of facet joints before and during halo fixation, while dynamic CT at neutral and maximally rotated head positions was performed to measure rotatory ROM after halo removal. The duration from onset to initial visit was significantly longer (3.2 vs. 5.7 months, *p* = 0.04), incidence of DOC during halo fixation was higher (0/7 [0%] vs. 4/5 [80%], *p* = 0.004), and segmental rotatory ROM of Occiput/C1 (Oc/C1) at final follow-up was larger (9.8 vs. 20.1 degrees, *p* = 0.003) in the fusion group. Long duration from the onset to the initial visit might induce irreversible damage to the articular surface of the affected facet, which was confirmed as DOC during halo fixation and resulted in SOF. Long duration from the onset to the initial visit and DOC during halo fixation could be used to suggest the risk for SOF. Nonetheless, rotatory ROM of Oc/C1 increased to compensate for SOF.

## 1. Introduction

Atlantoaxial rotatory fixation (AARF) is observed mainly in children. Patients suffer from neck pain and typically present a “cock-robin” position of the head and neck with limited range of motion (ROM) of the cervical spine. Acute AARF usually responds well to conservative treatment with a collar and traction if necessary. However, longer duration from the onset to the initial visit and delayed diagnosis can result in failure of conservative treatment and increase the requirement for more invasive treatment such as surgical reduction and fixation [1,2]. Fractures and congenital bone anomalies also need to be focused as a potential cause of AARF [3,4,5,6]. Accordingly, transition from acute to chronic or recurrent AARF needs to be prevented. Nonetheless, a recent systematic review concluded that there is a lack of high-quality data to determine the optimal nonsurgical treatment strategy [2].

The treatment strategy for chronic or recurrent AARF remains controversial. Surgical intervention used to be the mainstream treatment option for chronic AARF. Fielding and Hawkins [7] reported a series of 17 patients with chronic AARF and recommended surgery when the anterior displacement of the atlas is more than four millimeters. Parkih et al. [8] indicated surgical treatment when the duration from the onset to the initial visit is longer than three months. In subsequent clinical studies, several algorithms were advocated as the treatment of chronic AARF. Phillips and Hensinger [9] and Pang and Li [10] proposed halter or skull caliper tractions as an initial treatment, followed by halo fixation when reduction is achieved or surgical reduction and posterior fixation when the subluxation is irreducible or recurrent. Crossman et al. [11] proposed to start with a trial of halo traction to reduce subluxation. When reduced, the halo traction is converted to halo jacket fixation for three months. When not reduced, subluxation is surgically reduced followed by halo fixation, without surgical fixation of the atlantoaxial joint, to maintain the rotatory movement. However, surgical invasiveness and relatively high recurrence rate [9,10] were of concern in these treatment algorithms. Hence, even for chronic AARF, several studies have reported the efficacy of conservative treatments to reduce the invasiveness of the treatment, including rehabilitation [12], simple traction and bracing [13], halter traction and bracing [14,15,16,17], and closed manipulation under general anesthesia followed by external fixation [18,19,20,21,22,23].

We reported that chronic irreducible and recurrent unstable AARF can be managed successfully by “remodeling therapy,” a careful closed manipulation under general anesthesia followed by halo-vest fixation that obviates the need for surgical intervention [24,25]. Remodeling of C2 superior facet deformity is induced during halo fixation and can be detected on follow-up computed tomography (CT) images. It is a useful radiographic parameter to determine the appropriate period of halo fixation to prevent the recurrence of subluxation. However, we have experienced several cases in which occiput (Oc)-C1, C1-C2, or C2-C3 segment was osseously fused after remodeling therapy. It was hypothesized that there should be clinical (e.g., demographics of patients, duration from the onset, duration of halo fixation) and radiological findings (e.g., magnitude of sub-luxation, reduction during halo fixation) which suggest the risk of spontaneous osseous fusion (SOF). Interestingly, even when affected segments were osseously fused, rotatory range of motion (ROM) of the cervical spine recovered gradually after halo removal. The purpose of this study was to investigate the incidence and risk factors of SOF after remodeling therapy, and to reveal the mechanism to regain rotatory ROM of the cervical spine after the halo removal.

## 2. Materials and Methods

### 2.1. Patient Population

The subjects consisted of consecutive 23 patients with chronic AARF who underwent remodeling therapy for chronic AARF [19,24,25] at Keio University Hospital from 2009 to 2016. Following approval from the institutional review board, we performed a retrospective review of clinical records of 12 patients (three boys and nine girls) who received dynamic CT images at the initial visit and during follow-ups. All patients were referred to our hospital due to persistent torticollis which had been resistant to the primary conservative treatment for more than 1.5 months. They presented with neck pain and exhibited a so-called “cock-robin” posture with the head rotated to one side and flexed laterally to the other side in addition to restricted neck motion. The onset date of torticollis was provided by the referral letter from the hospital or clinic of origin, and was also confirmed by interview to the patient at our hospital.

### 2.2. Diagnosis and Grading of AARF

All patients underwent plain cervical radiography and CT scanning at the initial visit to exclude underlying fractures and congenital deformities [3,4,5,6]. The diagnosis of chronic AARF was established in all cases by confirming the deformity of the C2 superior facet (C2 facet deformity) on three dimensional CT reconstructions (3D-CT). The C1 lateral inclination angle was measured on 3D CT as we reported previously [24]. AARF was classified according to Fielding’s classification [7] and Ishii’s grading system [24]. Ishii’s grading system is based on the magnitude of both C2 facet deformity and C1 lateral inclination observed on 3D-CT: Grade I, AARF without C2 facet deformity or C1 lateral inclination; Grade II, with C2 facet deformity and C1 lateral inclination <20°; Grade III, with C2 facet deformity and C1 lateral inclination ≥20° [24].

### 2.3. Remodeling Therapy of AARF and Imaging Analyses

All patients underwent the following remodeling therapy: (1) careful closed manipulation and reduction of the atlantoaxial joint under general anesthesia; (2) halo-vest fixation to maintain reduction; (3) halo removal when the remodeling of C2 facet deformity was confirmed on 3D-CT; (4) gentle neck rotation exercise in supine position for approximately 10 min, three times per day [15,20,21]. During and after the remodeling therapy, follow-up CT scans were performed right after closed reduction/halo fixation to confirm the reduction and at two and three months (if necessary) after halo fixation to confirm the remodeling of the C2 facet deformity. Presence or absence of direct osseous contact (DOC) of facet joints and SOF of vertebrae were evaluated using sagittal and coronal CT images at the initial visit and during and after halo fixation. DOC was defined as disappearance of facet joint space between the adjacent vertebrae (Figure 1B and Figure 2B,D). At some time points after halo removal, all patients underwent dynamic CT examination, which was examined in three different positions (i.e., the neutral position and maximally rotated positions to both sides) as proposed by Pang and Li [26,27]. Using the final follow-up dynamic CT images, the angles made by Oc, C1, C2, C3, and C4 with the vertical line (rotatory vertebra angle) were recorded respectively as vertebrae angles (Oc~C4 angle) on the maximum rotated head position to each side [26,27]. Separation angles (subtraction of the rotatory vertebra angles) between adjacent vertebrae were calculated at each segment on each head position, and the sum of the separation angles in each head position was defined as the segmental rotatory range of motion (rotatory ROM: Oc/C1~C3/C4) [26,27]. For all CT scans, low-dose (one-fifth) radiation exposure was used in all 12 patients (Canon Medical Systems, Otawara, Japan. Aquilione ONE, e.g., setting of 2 mm slice thickness, 80 kV, 100 mA, CT Dose Index [CTDI] 2.60 mGy, with a 0.35-s scan time). In addition to the CT measurements, macroscopic rotatory ROM of the cervical spine was measured at the initial visit, at two weeks after halo removal, and at the final follow-up as previously reported [25]. Outward rotatory ROM of the cervical spine was also measured. The angle by the midline with the line connecting occiput and nose was measured from overhead on the maximum rotated head position to each side in a sitting position. The sum of these angles on each head position was defined as macroscopic rotatory ROM.

### 2.4. Study Groups

Five patients (5/12, 41.7%) whose images showed SOF of the cervical spine after remodeling therapy were grouped in the SOF group. The other 7 patients (7/12, 58.3%) without SOF during the follow-up period were grouped in the Non-SOF group. Clinical (i.e., age, sex, cause of AARF, duration from onset to initial visit, duration of halo fixation, macroscopic rotatory ROM) and radiographic data including plain radiographs and plain and 3D CT (atlantodental interval [ADI], C1 lateral inclination, Fielding’s classification, Ishii’s grading, DOC, SOF, segmental rotatory ROM) were retrospectively reviewed in both groups.

### 2.5. Statistical Analyses

Continuous variables were presented as mean ± standard deviation (SD). A comparison of each independent continuous variable (i.e., age, duration from onset to initial visit, C1 lateral inclination, duration of halo fixation, Macroscopic cervical ROM, segmental rotatory ROM, time from reduction to final CT) between the two groups was performed using an independent *t*-test. A paired *t*-test was used to analyze the change in macroscopic ROM of the cervical spine from two weeks after halo removal to the final follow-up. The Fisher’s exact test and the chi-square test for independence were used to analyze discrete variables (i.e., sex distribution, cause of AARF, Fielding’s classification, Ishii’s grading, presence or absence of DOC of facet joints). Statistical Package for the Social Sciences (SPSS, version 24; IBM Corp., Armonk, NY, USA) was used for all statistical analyses. Probability values of less than 0.05 were used to denote statistical significance. Post hoc power analyses were performed for the variables with statistical significance (*p* < 0.05) by G*Power (Franz Faul, Kiel University, Kiel, Germany. Version 3.1.9.6).

### 2.6. Compliance with Ethical Standards

We certified that all applicable institutional and governmental regulations concerning the ethical use of human participants and the ethical standards in the 1964 Declaration of Helsinki and its later amendments were followed during the research. Informed consent was obtained from all participants in the study. This study was approved by the Institutional Review Board of Keio University School of Medicine (IRB No. 20110142).

## 3. Results

### 3.1. Patient Characteristics and Incidence of Spontaneous Osseous Fusion

The characteristics of the 12 patients included are summarized in Table 1. The mean age at the initial visit was 7.6 ± 2.0 years (range 4.3–11.6). The mean duration from the onset of torticollis to the initial visit to our institute was 4.3 ± 2.2 months (range 1.5–8.8). The mean follow-up period was 6.3 ± 2.4 years (range 1.8–8.9). The most frequent cause of torticollis was upper respiratory tract infection (4/12, 33.3%), followed by an incidental minor trauma (2/12, 16.7%) and lymphangitis (2/12, 16.7%). Out of the 12 patients, spontaneous osseous fusions of adjacent vertebrae were confirmed in five patients (SOF group, 41.7%) after remodeling therapy: one in Oc/C1 (case 10), one in Oc/C1/C2 (case 8), two in C1/C2 (case 9 and 11) and one in C2/C3 (case 12). In all five patients, osseous fusions were confirmed on follow-up CT after halo removal, and not at the initial visit or during halo fixation.

### 3.2. Comparison of Clinical and Radiological Findings

The comparison of clinical and imaging variables between the two groups are summarized in Table 2. There were no significant differences in the mean age at the initial visit (non-SOF vs. SOF: 8.1 [range 7–9] vs. 7.3 [range 4–11], *p* = 0.489), sex distribution (male percentage, 2/5 [40%] vs. 1/7 [14.3%], *p* = 0.523), and the causes of chronic AARF (*p* = 0.334). In the SOF group, the mean duration from the onset of torticollis to the initial visit was significantly longer than that in the non-SOF group (3.2 ± 1.4 [range 1.5–6.0] vs. 5.7 ± 2.3 [range 3.4–8.8] months, *p* = 0.040, statistical power 0.52) (Table 2). The mean duration of halo fixation was similar in both groups (2.5 ± 0.4 vs. 2.9 ± 0.7 months, *p* = 0.221).

The radiological classification of AARF at the initial visit assessed by Fielding’s classification (*p* = 0.180) and Ishii’s grading (*p* = 0.079) was similar in both groups, although the incidence of Ishii grade III was likely to be higher in the SOF group compared to the non-SOF group. The mean C1 lateral inclination angles (non-SOF vs. SOF: 15.6 ± 13.9 vs. 21.8 ± 5.3, *p* = 0.369) and the mean atlantodental intervals (ADI, 5.0 ± 3.6 vs. 7.3 ± 2.5, *p* = 0.260) were larger in the SOF group but without statistical significance.

### 3.3. Direct Osseous Contact of the Facet Joint before and during Halo Fixation

In the non-SOF group, DOCs of facet joints at the initial visit were assessed in five patients excluding two patients whose 3D CTs were not obtained. DOCs were detected at the C1/C2 facet joint of the dislocated side in three out of five patients (60%) at the initial visit. However, DOCs disappeared after reduction and were not found in any patients during halo fixation (0%) (Table 1 and Table 2, and Figure 1).

In the SOF group, excluding one patient whose 3D CTs were not obtained (case 8), four out of four patients (100%) showed DOCs at the initial visit (three patients [case 9, 11, and 12] at the unilateral C1/C2 facet of the dislocated side and another patient [case 10] at the C1/C2 facet of the dislocated side and at the Oc/C1 facet of the contralateral side, Table 1). After reduction, DOCs disappeared in one patient (case 10), disappeared but another DOC was detected at the caudal segment in the contralateral side in one patient (case 12), and persisted at the same segment in two patients (case 9 and 11) (Figure 2). Therefore, four out of five patients (80%), including one patient whose 3D CT at the initial visit was not obtained (case 8), showed DOCs during halo fixation. Case 10 was the only case that showed spontaneous osseous fusion at the segment where DOC was found only at the initial visit and disappeared after reduction and during halo fixation.

The incidence of DOCs of facet joints was similar at the initial visit in both groups (non-SOF vs. SOF: 3/5 [60%] vs. 4/4 [100%], *p* = 0.151), which might be biased due to the exclusion of two patients in the non-SOF group and one patient in the SOF group due to the unavailability of 3D CTs. However, the incidence was significantly higher in the SOF group during halo fixation (0/7 [0%] vs. 4/5 [80%], *p* = 0.004, statistical power 1.0) (Table 2). The sensitivity of DOC during halo fixation to predict SOF after halo removal was 80% (four out of five), the specificity was 100% (seven out of seven), the positive predictive value was 100% (four out of four), and the negative predictive value was 87.5% (seven out of eight). The percentage of SOF of the segments with DOC during halo fixation was likely to be higher than that of the segments with DOC at the initial visit (4/4 [100%] vs. 3/7 [42.9%], *p* = 0.058) (Table 1).

### 3.4. Recovery of Rotatory ROM after Halo Removal

All 12 patients showed gradual recovery of rotatory ROM of the cervical spine after halo removal and had no impairment in activities of daily living. Macroscopic rotatory ROM of the cervical spine was significantly smaller in the SOF group than in the non-SOF group at all three time points (i.e., at the initial visit, two weeks after halo removal, and at the final follow-up) (Table 2). Statistical power was 0.74, 1.0, and 0.99, respectively. Patients in the NF group showed almost full recovery of macroscopic rotatory ROM [28,29,30] within two weeks after halo removal (173.0 ± 5.9), which was maintained at the final follow-up (174.3 ± 3.8, *p* = 0.235). In contrast, patients in the SOF group showed almost no recovery within two weeks after halo removal (57.0 ± 4.1), followed by gradual but significant recovery until the final follow-up (118.3 ± 26.3, *p* = 0.018).

The mean duration from reduction to the final follow-up dynamic CT was significantly longer in the SOF group (37.8 ± 15.0 months) than in the non-SOF group (8.3 ± 3.6 months, *p* = 0.011). The segmental rotatory ROM of C1/C2 in the non-SOF group was 50.7 ± 9.9 degrees at the final follow-up. As shown in Table 2, on the final follow-up of dynamic CTs, patients in the SOF group were likely to show larger rotatory ROMs at caudal adjacent C2/C3 (non-SOF vs. SOF: 7.9 ± 3.2 degrees vs. 13.0 ± 3.4, *p* = 0.051) and C3/C4 (12.0 ± 2.8 vs. 15.4 ± 3.4, *p* = 0.115) than those in the non-SOF group. The segmental rotatory ROM at rostrally adjacent Oc/C1 was significantly larger in the SOF group than in the non-SOF group (9.8 ± 3.5 vs. 20.1 ± 2.3, *p* = 0.003, statistical power 0.99).

## 4. Discussion

In the current study, five out of the twelve patients (41.7%) exhibited SOF of the affected segments after remodeling therapy for chronic AARF. Longer duration from the onset of torticollis to the initial visit and DOCs of the facet joints during halo fixation were identified as potential risk factors for SOF. Additionally, for the first time, we revealed that rotatory ROM at the adjacent segments of the fused segment, specifically at Oc/C1, increased and contributed to the recovery of rotatory ROM of the cervical spine.

### 4.1. Spontaneous Osseous Fusion after Halo Traction/Fixation

Previous studies have reported that long-term subluxation of C1/C2 can result in the deformity and osseous/fibrous union of the segment that prevents successful reduction [10,24,27,31]. However, it remained unknown whether SOF can also occur after conservative treatment for chronic AARF [13,17], including remodeling therapy. Dove et al. [32] reported an SOF of the cervical spine as a complication after halo-pelvic traction for stiff spinal deformities (e.g., a tuberculous kyphosis and an idiopathic scoliosis). The duration of halo-pelvic traction was longer than 3 months, and most of the patients were older than 11 years. The fusion segments were found between C2 and C5. The authors hypothesized that the distraction force might have resulted in the disturbance of bony attachments of the cervical ligaments, causing local hemorrhage and later bone formation. This hypothesis may explain one potential mechanism of SOF induced by traumatic damage, though the ages of the patients, etiology, and duration of halo fixation were different from those in the current study. Krengel et al. [33] reported an SOF between Oc and C2 after closed reduction and halo fixation for chronic AARF in an 11-year-old boy. The duration from the onset to the initial visit was 6 months, and the duration of halo fixation was 3 months, which were similar to our study. Consistent with our series, the osseous fusion was confirmed after halo removal. The patient was found to be HLA-B27 positive, and the spontaneous fusion of the cervical spine has been reported to often occur in ankylosing spondyloarthropathy [34]. Accordingly, these authors suggested the possibility of pre-existing inflammatory condition caused by HLA-B27-related spondyloarthropathy, as well as ankylosis of C1/C2 at the initial visit and possible trauma to the joint during closed reduction, could have contributed to the development of osseous fusion.

In the current study, five out of the twelve patients (41.7%) exhibited SOF. Long duration from the onset to the initial visit and DOCs of the facet joints during halo fixation were identified as potential risk factors for SOF. DOC, which is the disappearance of joint space, may indicate damage to the articular cartilage. Damage to the articular cartilage leading to osteoarthritis of the affected facet joints may consequently result in SOF [11]. All patients in the non-SOF group showed DOC only at the initial visit, and DOC disappeared after reduction and during halo fixation (Figure 1, Table 1). On the other hand, in the SOF group, DOC at the initial visit persisted during halo fixation, even after subluxation was well reduced (Figure 2, Table 1). As shown in Case 10 (Table 1), the segment with DOC only at the initial visit also resulted in SOF, unlike the non-SOF group. Additionally, in terms of the recovery pattern of the macroscopic rotatory ROM of the cervical spine, the two groups showed obviously different processes after similar duration of halo fixation. The rotatory ROM was already significantly smaller in the SOF group than in the non-SOF group at the initial visit. Furthermore, patients in the SOF group showed almost no recovery at two weeks after halo removal, whereas those in the non-SOF group showed almost full recovery at the same time point (Table 2). These radiological and clinical findings seem to reflect one aspect of the severity of subluxation, which well-validated measures (i.e., Fielding’s classification and Ishii’s grading) cannot fully reflect due to the complexity in the pathophysiology of chronic AARF. It might be better to classify the severity of chronic AARF more comprehensively from the clinical as well as the morphologic perspective, as with the AO Spine Subaxial Cervical Spine Injury Classification System [35]. Therefore, we presume that DOC during halo fixation, limited rotatory ROM after halo removal, and resulting SOF might be dependent not on halo fixation itself, but on the severity of the damage of the affected facet joints that was already developed at the time of the initial visit. Significantly longer duration from the onset to the initial visit could have damaged the joint cartilage more severely in the SOF group than in the non-SOF group. It is hypothesized that, when the damage of the facet joint is already irreversible at the initial visit, DOC is likely to persist even after successful reduction and result in joint contracture and SOF. However, it was difficult to evaluate the damage using our CT images because we tried to minimize the radiation exposure using modalities just sufficient to recognize the obvious bony landmarks. Precise evaluation of the severity and area of the damage to joint cartilage is required to clarify this hypothesis. Magnetic resonance imaging (MRI) can be a safer alternative method regarding the issue of radiation exposure. Additionally, MRI is useful to evaluate the extent of inflammation at the atlantoaxial joint and its surrounding anatomical structures (e.g., retropharyngeal space, paranasal sinuses).

We previously revealed that a prominent lateral inclination of C1 is an impeding factor for reduction of chronic AARF and indicates an irreducible subluxation when greater than 20° [24]. Crockard, et al. reported that reduction is most likely to be prevented by the development of abundant fibrous tissue at C1/C2 [10,27,31]. Ishii’s grading also explains the severity of AARF according to a time-dependent process from the acute to the chronic phase of AARF [24]. Although there were no significant differences in the incidence of Ishii type 3 and C1 lateral inclination between the two groups at the initial visit, a significantly longer duration from the onset to the initial visit might have made the reduction more difficult in the SOF group, requiring over-correction of the atlas to the contralateral side, which might have induced another DOC on the contralateral side at the caudal adjacent segment, as seen in Case 12 (Table 1). Adjustment of the head position may be considered when a new DOC is found at the adjacent segment on the 3D-CT immediately after reduction.

### 4.2. Compensatory Mechanism to Gain Rotatory ROM after Halo Removal

A previous study reported that patients with chronic AARF showed reduced rotatory ROM of the cervical spine after open reduction and halo fixation but regained some rotatory ROM [11]. We reported that rotatory ROM of the cervical spine was limited after halo removal when C1/C2 fibrous contracture developed or the articular surface of the C1/C2 facet was damaged by traumatic events, but gradually improved during the follow-up [24,25]. However, the mechanism of the recovery remained unclear.

In the current study, it is noteworthy that patients in the SOF group showed gradual but significant recovery of rotatory ROM of the cervical spine until the final follow-up. Interestingly, those patients showed significantly increased segmental rotatory ROM at Oc/C1 and were likely to show larger rotatory ROM at the caudal adjacent segments (C2/C3 and C3/C4) than those in the non-SOF group at the final follow-up (Table 2). In contrast to the atlantoaxial joint which allows movements in all directions [36], the major motion at Oc/C1 is considered flexion and extension and the movement of rotation is a coupled motion [37], because of the cradle structure of the Oc/C1 joint and the lack of direct muscle involvement in the rotation [38]. The reported rotatory ROMs at the Oc/C1 segment to each side (i.e., right or left) are similar, ranging from 1.7° to 3.9°, measured by an MRI [20] or the CT [39,40,41] 3D reconstruction method and dual fluoroscopic imaging and model registration techniques [42,43] in healthy adults, and less than 3° in normal children younger than 12 years [26].Compared to these values in normal subjects, this study revealed markedly increased rotatory ROM of Oc/C1 in the SOF group (20.1 ± 2.3) as well as in the non-SOF group (9.8 ± 3.5). These findings suggest the possibility that rotatory ROM at adjacent segments increased to compensate for the C1/C2 segment, especially when the segment was osseously fused. Sternocleidomastoid (SCM) inserts into the mastoid process of the skull to allow for greater moment arm for head rotation than with rotator cervicis within the cervical spine, and work as a major rotator of the head on the cervical spine. This anatomical consideration might explain why the compensatory increase in the segmental rotatory ROM was predominant at rostrally adjacent Oc/C1 than caudally adjacent segments. Additionally, duration from reduction to the final follow-up of dynamic CT was significantly longer in the SOF group (27.7 ± 6.7 months) than in the non-SOF group (8.3 ± 3.6 months). We presume that rotatory ROM of Oc/C1, as well as caudally adjacent segments, might increase over time after halo removal to compensate for the affected C1/C2 joint as the natural history, regardless of the presence or absence of SOF.

### 4.3. Limitations

The present study has several limitations. First, this study is a retrospective comparative study that included only a small number of patients and can carry the risk of sample bias. The statistical power of the independent t test for the duration from onset to initial visit was small (0.52). Multivariate logistic regression analysis was not performed to investigate the independent predictors of SOF because of the small sample size. Univariate analysis was not performed because the incidence of DOC during halo fixation was 0% in the non-SOF group, and the odds ratio could not be calculated. Second, the dynamic behavior of the segments below C4 down to the thoracic spine after remodeling therapy remains unclear. Their contribution to the recovery of rotatory ROM of the cervical spine needs to be examined. Third, long term follow-up by dynamic CT is desired to examine the changes in the rotatory ROM at each segment as natural history after the remodeling therapy, and to identify the timing when the osseous fusion is completed. Due to the issue of radiation exposure, we had to limit the frequency and area of CT scans. Additionally, long term effects of the compensation in the ROMs of proximally and distally adjacent segments should be investigated, because the adjacent segments may result in osteoarthritis at earlier time point. Fourth, values of segmental rotatory ROMs of the cervical spine in healthy subjects were not examined in this study. These values may clarify the compensatory mechanism to regain rotatory ROM by assessing the changes in rotatory ROM at each segment after the remodeling therapy.

## 5. Conclusions

To the best of our knowledge, this is the first study to report the incidence of SOF after the remodeling therapy for chronic AARF, its risk factors, and a possible mechanism to regain rotatory ROM of the cervical spine after halo removal. Long duration from the onset to the initial visit and DOC during halo fixation could be used to suggest the risk for SOF. DOC during halo fixation, limited rotatory ROM after halo removal, and resulting SOF might be dependent not on halo fixation itself, but on the severity of the damage of the affected facet joints which had already developed at the time of the initial visit. The primary purpose of remodeling therapy is to prevent the recurrence of subluxation without fixing the C1/C2 joint thereby preserving the C1/C2 rotatory motion. Even if unfavorable SOF occurs after the remodeling therapy, recurrence of subluxation, the primary purpose, can still be achieved without causing any surgical impact to the patients. Even in such a case, limited rotatory ROM is compensated, at least in part, by increased motion in the adjacent segments, especially the rostral Oc/C1.

This study provides clinical insights that will help surgeons to understand the dynamic behavior of the cervical spine after remodeling therapy and to select a less invasive treatment for chronic AARF, obviating surgical invasiveness.

## Figures and Tables

**Figure 1 jcm-11-01504-f001:**
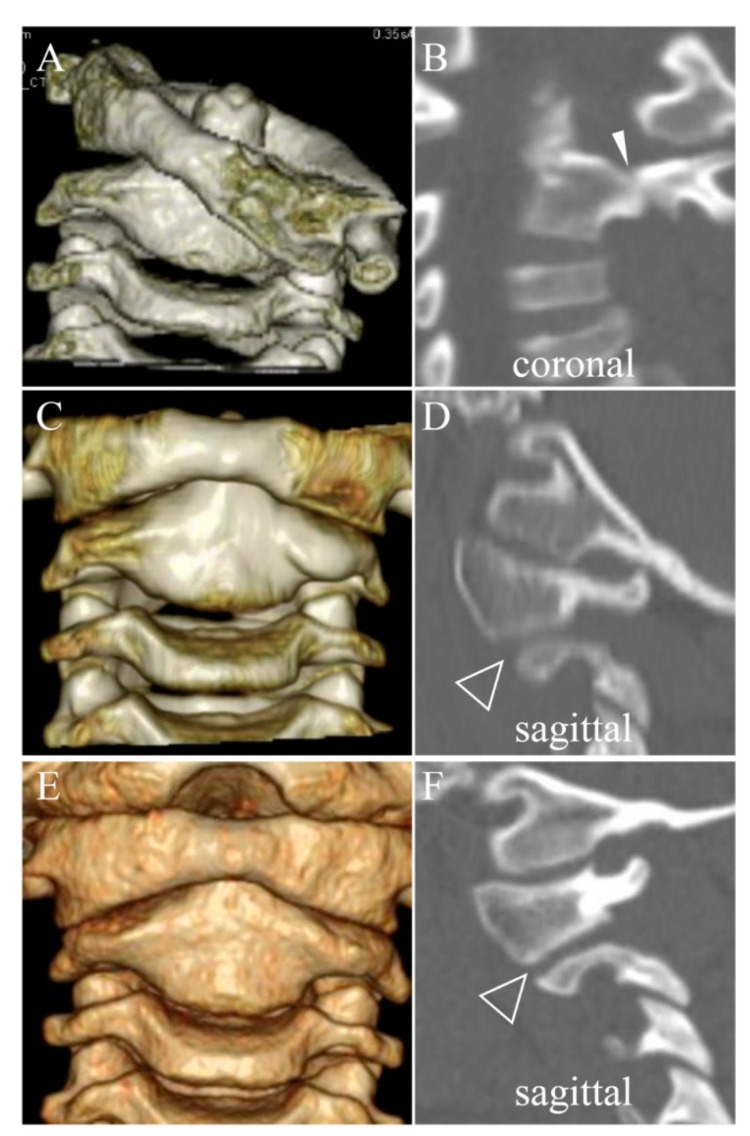
Representative non-fusion case (case 7) of 6-year-pld boy with Fielding type 3 and Ishii grade 3 (**A**). Direct osseous contact of C1/C2 facet on the dislocated side at initial visit (solid arrowhead in (**B**)), disappeared after reduction and during halo fixation (**C**), blank arrowhead in (**D**), and resulted in remodeling of facet deformity without osseous fusion at the final follow-up (**E**), open arrowhead in (**F**). Duration of halo fixation was 2.6 months.

**Figure 2 jcm-11-01504-f002:**
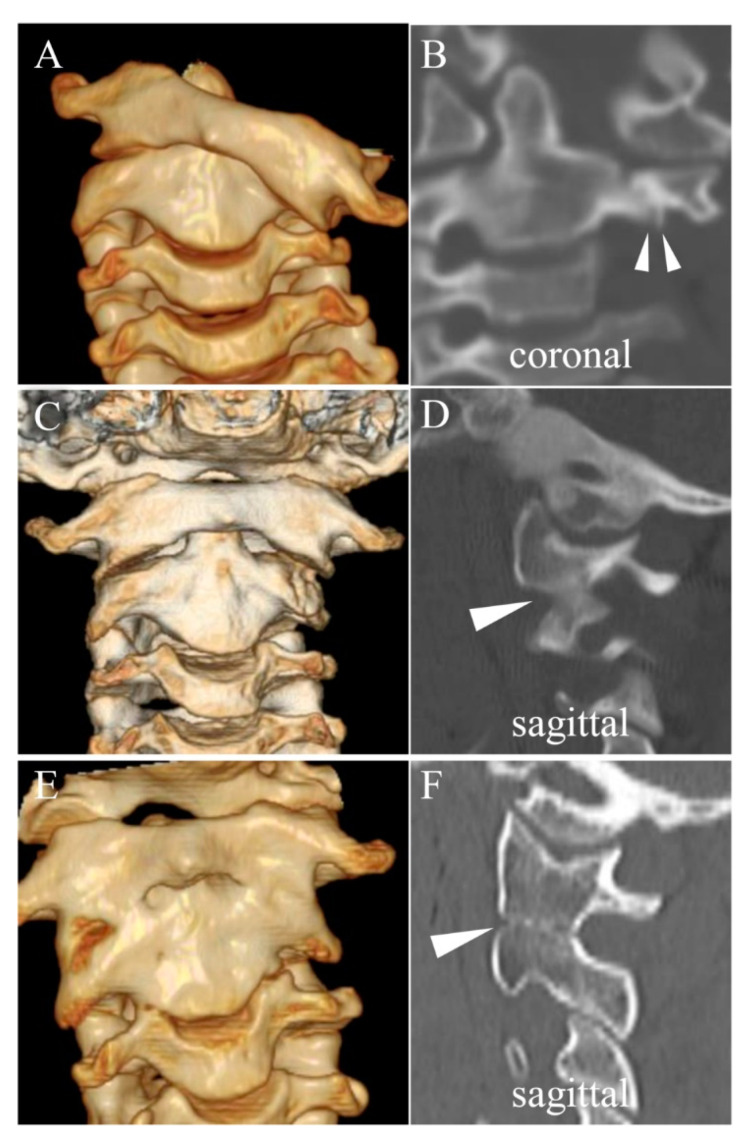
Representative osseous fusion case (Case 9) 7-year-old female patient with Fielding type 3 and Ishii grade 3 (**A**). Direct osseous contact of C1/C2 facet on the dislocated side at initial visit (solid arrowhead in (**B**)), persisted after reduction and during halo fixation (**C**), solid arrowhead in (**D**), and resulted in spontaneous osseous fusion at the final follow-up (**E**), solid arrowhead in (**F**). Duration of halo fixation was 2.9 months.

**Table 1 jcm-11-01504-t001:** Patient Characteristics.

	Case	Age	Sex	Cause of AARF	Duration from Onset to Initial Visit (Months)	FC	IG	ADI (mm)	C1 LI (Degree)	Duration of Halo Fixation (Months)	Direct Osseous Contact of Facet		Osseously Fused Segment after Halo Removal
											Initial Visit	During Halo Fixation	
Non-SOF group	1	11	F	Upper RTI	6	3	2	13	18	3.1	Unknown *	-	-
	2	8	F	Unknown	3.8	2	2	3.6	18	2.3	Unknown *	-	-
	3	4	F	Upper RTI	3.4	2	3	4.2	38	2.1	C1/C2	-	-
	4	5	F	Upper RTI	2.1	1	2	3.5	−3	2.1	-	-	-
	5	5	F	Otitis media	2.6	2	2	5	4	3	-	-	-
	6	9	F	Lymphangitis	1.5	1	2	2	8.5	2.1	C1/C2	-	-
	7	6	F	Kawasaki disease	3.2	3	3	4	26	2.6	C1/C2	-	-
SOF group	8	7	F	Upper RTI	5.4	3	3	7	28	3.8	Unknown *	Oc/C1, C1/C2	Oc-C2
	9	7	F	Minor trauma	8.8	3	3	8	24	2.9	C1/C2	C1/C2	C1-C2
	10	9	M	Minor trauma	3.9	3	2	10.2	13.5	2.8	Oc/C1, C1/C2	-	Oc-C2
	11	8	F	Mumps	7.2	3	3	3.3	22.8	1.9	C1/C2	C1/C2	C1-C2
	12	7	M	Lymphangitis	3.4	2	3	8	20.9	3	C1/C2	C2/C3	C2-C3

* Presence or absence of direct osseous contact of facet joint at initial visit was unknown in case 1, 2, and 8 because sagittal and coronal 3D reconstructions were not obtained. FC: Fielding Classification; IG: Ishii Grading; ADI: atlantodental intervals; C1 LI: C1 lateral inclination. RTI: respiratory tract infection.

**Table 2 jcm-11-01504-t002:** Comparison of clinical and imaging variables.

Parameter	Non-SOF Group	SOF Group	*p* Value
	Case 1–7	Case 8–12	
No. of patients	7	5	
Age (yrs)			
mean	7.3 ± 2.6	8.1 ± 0.7	0.489
range	4–11	7–9	
Sex			
male	1	2	
female	6	3	0.523
Duration from onset to initial visit (months)	3.2 ± 1.4	5.7 ± 2.3	0.040
Causes (no. of cases)			
upper RTI	3	1	
minor trauma	0	2	
lymphangitis	1	1	
otitis media	1	0	
Kawasaki disease	1	0	
mumpus	0	1	
unknown	1	0	0.334
Fielding classification (no. of cases)			
I	2	0	
II	3	1	
III	2	4	0.180
Ishii grading (no. of cases)			
I	0	0	
II	5	1	
III	2	4	0.079
C1 lateral inclination (degree)	15.6 ± 13.9	21.8 ± 5.3	0.369
ADI (mm)	5.0 ± 3.6	7.3 ± 2.5	0.260
Macroscopic cervical rotatory ROM (degree)			
At initial visit	66.5 ± 7.7	38.0 ± 22.2	0.018
Two weeks after halo removal	173.0 ± 5.9	57.0 ± 4.1	<0.001
At final follow-up	174.3 ± 3.8	118.3 ± 26.3	<0.001
Duration of halo fixation (months)	2.5 ± 0.4	2.9 ± 0.7	0.221
Direct osseous contact of facet joints (no. of cases)			
At initial visit * Present	3	4	
Absent	2	0	0.151
During halo fixation Present	0	4	
Absent	7	1	0.004
Segmental rotatory ROM on final follow-up CT			
Oc/C1	9.8 ± 3.5	20.1 ± 2.3 ^†^	0.003
C1/C2	50.7 ± 9.9	-	-
C2/C3	7.9 ± 3.2	13.0 ± 3.4 ^‡^	0.051
C3/C4	12.0 ± 2.8	15.4 ± 3.4	0.115
Time from reduction to final CT (mos)	8.3 ± 3.6	37.8 ± 14.9	0.011

* Two patients in the non-SOF group (Case 1, 2) and one patient in the SOF group (Case 8) were excluded because their 3D-CT reconstructions at initial visit were not obtained. ^†^ Case 8 and Case 10 were excluded from the calculation because Oc/C1 segment was fused. ^‡^ Case 12 was excluded from the calculation because C2/C3 segment was fused. RTI: respiratory tract infection.

## Data Availability

The data presented in this study are available on request from the corresponding author. The data are not publicly available due to ethical and privacy restrictions.

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
