# Peer review of "Spontaneous Osseous Fusion after Remodeling Therapy for Chronic Atlantoaxial Rotatory Fixation and Recovery Mechanism of Rotatory Range of Motion of the Cervical Spine"

_jcm, 2022, doi:10.3390/jcm11061504_

Round 1
Reviewer 1 Report
This topic is interesting, please look at these points:
- Lines 63-65: "When not reduced, open reduction is performed followed by halo fixation but not surgical fixation, to maintain the rotatory movement of atlantoaxial joint" What do authors mean with "open reduction" without surgery ? This sentence is not correct, revise.
- Lines 48-50 and Lines 96-97. It is important to highlight the attention between fracture and congenital bone anomalies. Ref. Congenital absence of a cervical spine pedicle. Neurol India. 2016 Jan-Feb;64(1):189-90. doi: 10.4103/0028-3886.173669. --- Congenital absence of the neural arch. Eur J Radiol. 1995 Jul;20(2):100. --- Congenital absence of a cervical spine pedicle: clinical and radiologic findings. AJR Am J Roentgenol. 1990 Nov;155(5):1037-41. --- Congenital anomalies of the cervical spine. Neuroimaging Clin N Am. 1995 Aug;5(3):427-49.
- Lines 86-87: "The subjects consisted of 23 patients with chronic AARF". Are these patients consecutive patients? If yes, state it.
- Lines 153-154: State here that patients included in the study are 12, than report mean age and other points.
- Lines 302-305: "these radiological and clinical findings seem to reflect one aspect of the severity of subluxation which well-validated measures .." Discuss about severity of cervical spine trauma, considering this AOSpine classification: Establishing the Injury Severity of Subaxial Cervical Spine Trauma: Validating the Hierarchical Nature of the AO Spine. Spine (Phila Pa 1976). 2021 May 15;46(10):649-657. doi: 10.1097/BRS.0000000000003873.
- Congratulations for figures 1 and 2
- Lines 343-345: "Pang et al [22] examined CT images in various head positions during head rotation and reported that Oc/C1 separation angle was less than 3 degrees to each side in normal chil345 dren, suggesting that rotatory ROM of Oc/C1 should be less than 6 degrees." These biomechanical concepts should be discussed more, similar considerations were reported also by Mangini & Papini, by Bogduk & Mercer and by Yanni & Perin.
- Lines 380-390: Conclusion: what could be the clinical value of this work?
Reviewer 2 Report
In this study, the authors investigated spontaneous osseous fusion (SOF) and range of motion (ROM) after the treatment of atlantoaxial rotatory fixation (AARF) using the conservative therapy of halo fixation. I appreciate that this study is clinically significant for the treatment of AARF, and may be attractive to clinical practitioners. However, my major concern is that the outcomes of only 12 patients were assessed, so it is hard to convince me of the statistical power in the hypothesis tests of risk factors, such as pre-treatment direct osseous contact (DOC) of facet joints and the duration from the onset of AARF to the initial visit. My detailed comments were listed below for the authors to consider and address:
TITLE:
It is fine but a little lengthy.
ABSTRACT:
Throughout this paper, fusion refers to the disorder of SOF, rather than the surgery of spinal fusion. To distinguish from spinal fusion, can the authors emphasize the fusion/non-fusion groups to be the SOF/non-SOF groups?
P1, L27: please move the abbreviation of “DOC” to the accurate location.
It is unclear that the paired values in the parentheses correspond to either the SOF group or the non-SOF group? Please specify these values, e.g., “non-SOF vs SOF: 3.2 vs 5.7 months, P=0.04” (please double-check if I correctly mark these two groups).
In this paper, the terminology of surgical approaches and pathologies may confound the readers. Can the authors consider distinguishing them by providing a list?
INTRODUCTION:
P2, L69: Is “remodeling therapy” special terminology for halo fixation?
P2, L77-80: For the objective of this study, (statistical) hypotheses tested in this study should be introduced here.
MATERIALS AND METHODS:
P2, L86: Are 12 patients sufficient for statistical analysis? I suggest that a statistical power analysis (at least, post hoc) regarding variables with significance (p < 0.05) should be added and reported.
P3, L103-127: The manufacturers and modalities of medical imaging systems should be provided.
P3, L115: for dynamic CT examination, what was the position of patients during CT scanning?
ROM measurements should be carefully described. For example, how to define the coordinate systems to measure ROM? Further, did the author validate the accuracy and repeatability?
P3, L119: what is the meaning of “the vertical 0 degrees”?
P3, L124: what is “macroscopic rotatory ROM”?
P3, L128-133: Suggest changing the “fusion and non-fusion groups” to the “SOF and non-SOF groups” throughout the paper.
P3, L132: please specify what are the “Clinical and radiographic data”.
P3, L134-142: please further describe what are the independent and dependent variables in statistical analysis.
RESULTS:
P4, L170-171: Was this duration reported by the patients? Was it accurate? Please clarify it.
P6, L190: why “in three out of five patients”?
P7, L216: For “The incidence of DOCs of facet joints”, I feel that there are large biases, due to the exclusion and unavailability of patients' CT.
P7, L230: To measure the “full recovery of macroscopic rotatory ROM”, what was the reference (the normal ROMs) used?
DISCUSSION
P8, L255: Were your patients with chronic AARF? How was this diagnosed?
P8, L274: As mentioned before, this finding regarding the incidence of SOF may be highly biased.
P9, L292-294: I feel confused how this was proved by your data?
P9, L297-299: Hypotheses should be first proposed in the Introduction.
P9, L299-230: It is not clear why MRI should be used?
P9, L308: for C1 lateral inclination, Did the authors report this in the Results?
P9, L325-326: Except for these early studies using CT scanning in the supine position, recent investigations of upper cervical kinematics during functional neck axial rotation using other techniques in multiple groups should be referenced. Please see:
Anderst et al, 2017. Dynamic in vivo 3D atlantoaxial spine kinematics during upright rotation. J. Biomech. 60, 110–115. https://doi.org/10.1016/j.jbiomech.2017.06.007.
Kang et al, 2019. In vivo three-dimensional kinematics of the cervical spine during maximal active head rotation. PLoS One 14, 1–16. https://doi.org/10.1371/journal.pone.0215357.
Tang et al. In vivo 3-Dimensional Kinematics Study of the Healthy Cervical Spine Based on CBCT Combined with 3D-3D Registration Technology. Spine (Phila Pa 1976). 2021 Dec 15;46(24):E1301-E1310. doi: 10.1097/BRS.0000000000004231
Zhou et al. Intervertebral range of motion characteristics of normal cervical spinal segments (C0-T1) during in vivo neck motions. J Biomech. 2020 Jan 2;98:109418. doi: 10.1016/j.jbiomech.2019.109418
Guo et al. In vivo primary and coupled segmental motions of the healthy female head-neck complex during dynamic head axial rotation. J Biomech. 2021 Jun 23;123:110513. doi: 10.1016/j.jbiomech.2021.110513
P10, L346: In addition to these limitations, I think that long-term effects of the compensation in the ROMs of proximally and distally adjacent segments should be investigated.
Round 2
Reviewer 1 Report
Authors solved all my criticisms.
Author Response
Dear Reviewer #1,
Thank you again for your careful review and accepting our response to your comments. We are glad to know that our amendment reflected your suggestions correctly and answered your question.
Reviewer 2 Report
I highly appreciate that the authors have well responded to my comments and addressed my concerns.
Author Response
Dear Reviewer #2,
Thank you again for your careful review and accepting our response to your comments. We are glad to know that our amendment reflected your suggestions correctly and answered your question.
This manuscript is a resubmission of an earlier submission. The following is a list of the peer review reports and author responses from that submission.
Round 1
Reviewer 1 Report
This topic is interesting, please look at these points:
- Lines 63-65: "When not reduced, open reduction is performed followed by halo fixation but not surgical fixation, to maintain the rotatory movement of atlantoaxial joint" What do authors mean with "open reduction" without surgery ? This sentence is not correct, revise.
- Lines 48-50 and Lines 96-97. It is important to highlight the attention between fracture and congenital bone anomalies. Ref. Congenital absence of a cervical spine pedicle. Neurol India. 2016 Jan-Feb;64(1):189-90. doi: 10.4103/0028-3886.173669. --- Congenital absence of the neural arch. Eur J Radiol. 1995 Jul;20(2):100. --- Congenital absence of a cervical spine pedicle: clinical and radiologic findings. AJR Am J Roentgenol. 1990 Nov;155(5):1037-41. --- Congenital anomalies of the cervical spine. Neuroimaging Clin N Am. 1995 Aug;5(3):427-49.
- Lines 86-87: "The subjects consisted of 23 patients with chronic AARF". Are these patients consecutive patients? If yes, state it.
- Lines 153-154: State here that patients included in the study are 12, than report mean age and other points.
- Lines 302-305: "these radiological and clinical findings seem to reflect one aspect of the severity of subluxation which well-validated measures .." Discuss about severity of cervical spine trauma, considering this AOSpine classification: Establishing the Injury Severity of Subaxial Cervical Spine Trauma: Validating the Hierarchical Nature of the AO Spine. Spine (Phila Pa 1976). 2021 May 15;46(10):649-657. doi: 10.1097/BRS.0000000000003873.
- Congratulations for figures 1 and 2
- Lines 343-345: "Pang et al [22] examined CT images in various head positions during head rotation and reported that Oc/C1 separation angle was less than 3 degrees to each side in normal chil345 dren, suggesting that rotatory ROM of Oc/C1 should be less than 6 degrees." These biomechanical concepts should be discussed more, similar considerations were reported also by Mangini & Papini, by Bogduk & Mercer and by Yanni & Perin.
- Lines 380-390: Conclusion: what could be the clinical value of this work?